# Exploring the Association between Individual-Level Attributes and Fidelity to a Vocational Rehabilitation Intervention within a Randomised Controlled Trial

**DOI:** 10.3390/ijerph20064694

**Published:** 2023-03-07

**Authors:** Katie E. Powers, Roshan das Nair, Julie Phillips, Amanda Farrin, Kathryn A. Radford

**Affiliations:** 1Injury, Inflammation & Recovery Sciences, Queen’s Medical Centre, University of Nottingham, Nottingham NG7 2UH, UK; 2Health Division, SINTEF, 7465 Trondheim, Norway; 3Mental Health & Clinical Neurosciences, School of Medicine, University of Nottingham, Nottingham NG7 2UH, UK; 4Clinical Trials Research Unit, Leeds Institute of Clinical Trials Research, University of Leeds, Leeds LS2 9JT, UK

**Keywords:** complex intervention, implementation fidelity, vocational rehabilitation, stroke

## Abstract

Understanding what attributes or characteristics of those delivering interventions affect intervention fidelity and patient outcomes is important for contextualising intervention effectiveness. It may also inform implementation of interventions in future research and clinical practice. This study aimed to explore the relationships between attributes of Occupational Therapists (OTs), their faithful delivery of an early stroke specialist vocational rehabilitation intervention (ESSVR), and stroke survivor return-to-work (RTW) outcomes. Thirty-nine OTs were surveyed about their experience and knowledge of stroke and vocational rehabilitation and were trained to deliver ESSVR. ESSVR was delivered across 16 sites in England and Wales between February 2018 and November 2021. OTs received monthly mentoring to support ESSVR delivery. The amount of mentoring each OT received was recorded in OT mentoring records. Fidelity was assessed using an intervention component checklist completed using retrospective case review of one randomly selected participant per OT. Linear and logistic regression analyses explored relationships between OT attributes, fidelity, and stroke survivor RTW outcome. Fidelity scores ranged from 30.8 to 100% (Mean: 78.8%, SD: 19.2%). Only OT engagement in mentoring was significantly associated with fidelity (b = 0.29, 95% CI = 0.05–0.53, *p* < 0.05). Increased fidelity (OR = 1.06, 95% CI = 1.01–1.1, *p* = 0.01) and increasing years of stroke rehabilitation experience (OR = 1.17, 95% CI = 1.02–1.35) was significantly associated with positive stroke survivor RTW outcomes. Findings of this study suggest that mentoring OTs may increase fidelity of delivery of ESSVR, which may also be associated with positive stroke survivor return-to-work outcomes. The results also suggest that OTs with more experience of stroke rehabilitation may be able to support stroke survivors to RTW more effectively. Upskilling OTs to deliver complex interventions, such as ESSVR, in clinical trials may require mentoring support in addition to training to ensure fidelity.

## 1. Introduction

Results of intervention studies typically only report whether an intervention is shown to attain target outcomes, but attention is not often afforded to more nuanced considerations around intervention implementation [1,2,3]. Understanding the implementation of an intervention can provide helpful information about why or how an intervention did or did not attain the targeted outcomes [4,5,6]. This is especially true of ‘complex’ interventions which comprise numerous components, require certain expertise, skills, and behaviours of those delivering and receiving the intervention, and target multiple groups and settings [7,8]. A complex intervention often requires its components to be modified to fit the needs of those delivering or receiving the intervention [7,8], in these instances it is important to consider ‘fidelity’ or the extent to which an intervention is delivered [9,10,11,12].

Fidelity assessment can be useful not just as an implementation technique to monitor and support healthcare professionals delivering an intervention over the course of the study period [13,14,15], but also to help researchers contextualise the results of a study. Despite the importance of measuring fidelity, systematic reviews have shown that fidelity assessments are rarely conducted [16,17,18], or in cases where assessment has occurred, reported [19,20]. Higher levels of fidelity in studies of complex behaviour change interventions are linked to better patient outcomes [21,22,23,24], which suggests it is important for researchers to understand what factors promote fidelity. Achieving fidelity in some complex intervention studies can be difficult and studies within implementation research have therefore highlighted the importance for thorough investigation into factors which may influence fidelity outcomes [25,26,27]. Due to the number of stakeholders involved in the delivery and receipt of complex interventions, the influence of individual-level attributes on fidelity outcomes is of particular interest [14,28].

Individual-level attributes are personal characteristics that can be stable, i.e., thought to stay the same over time (e.g., education, openness to change, critical thinking) [29] or unstable, which are subject to change over time (practice, engagement, illness) [30]. Studies of the factors affecting the implementation of evidence-based practice (EBP) have shown that attributes of both healthcare professionals and patients may contribute to poor fidelity outcomes, or implementation failure [31,32,33]. Research regarding what therapist attributes (e.g., gender, years of experience, level of education, etc.) impact fidelity outcomes is inconsistent and inconclusive [14,34,35]. For example, while years since qualifying as a therapist does not demonstrate clear directionality [14,34,35], experience in skills related to an intervention improves fidelity rates [36]. Therapists’ positive attitudes and beliefs towards EPB [28,37], greater competency [38,39], and greater self-efficacy [40,41] have been found to consistently, considerably, and positively affect either fidelity or patient outcome across patient groups, intervention type, and setting [42,43,44]. Most studies exploring the impact of therapist attribute on fidelity outcomes are within studies of interventions delivered by psychotherapists (as detailed in systematic reviews) [43,44]. Research within complex intervention studies delivered by other professions outside of psychotherapy is lacking and inconclusive.

Vocational rehabilitation (VR) is an intervention that supports people in returning to or remaining in work [45]. Many long-term neurological conditions, including stroke, can affect a person’s ability to stay in or return-to-work. VR interventions are considered complex because they contain many interacting parts that are influenced by many different contexts and factors [46]. The delivery of VR requires the intervention to be tailored to the person receiving it, being sensitive to the changing behaviours of the different stakeholders or those delivering, receiving, and affected by the intervention. It crosses organisational boundaries and can produce a variety of outcomes [32]. Occupational therapists are healthcare professionals who support people’s activities of daily living and are therefore well-situated to deliver VR. VR interventions have been studied in stroke survivor populations [47,48], but there is no reported measurement of implementation considerations, such as fidelity or the individual-level attributes, that might be affect implementation or outcomes. Without information regarding the context around intervention delivery, confident conclusions regarding study results and intervention effectiveness are not possible, and the likelihood of patients receiving potentially life-changing intervention is diminished.

This study sought to explore the relationships between OT attributes, implementation fidelity, and stroke survivor return-to-work outcomes and was conducted using data collected from the RETurn to work After stroKE (RETAKE) trial (ISRCTN: 12464275); a large multi-centre randomised controlled trial (RCT) of a complex, VR intervention to support stroke survivors to return to and stay in work following their stroke.

## 2. Materials and Methods

We used a correlational design drawing on qualitative and quantitative data from RETAKE OTs and stroke survivors in the RETAKE trial intervention group to explore the relationships between OT attributes, implementation fidelity, and stroke survivor RTW outcome at 12-month post-randomisation in the RETAKE trial (Trial Registration: ISRCTN, ISRCTN12464275. Registered on 13 March 2018). Ethical approval for the RETAKE trial and the studies within the trial was obtained through the East Midlands–Nottingham 2 Research Ethics Committee (REC) (Ref: 18/EM/0019).

### 2.1. Participants

We recruited at least two OTs from each of RETAKE’s 16 study sites across England and Wales to be trained to deliver the intervention. Aside from being a qualified OT, there were no other inclusion/exclusion criteria for OT recruitment, but previous experience of stroke and VR in community settings was desirable.

Stroke survivors who were recruited to the RETAKE study [49], and whose data were included in this study, were required to be:Aged 18 years or older;Admitted to a hospital with a new stroke prior to recruitment;Working at the time of their stroke (paid or unpaid, for at least two hours per week);Have the capacity to provide informed consent for participation;Sufficient English language proficiency to participate in the study.

Stroke survivors who did not intend to return-to-work were excluded.

### 2.2. Early Stroke Specialist Vocational Rehabilitation

Early stroke specialist vocational rehabilitation (ESSVR) combines conventional VR with case management. It is delivered by a stroke specialist OT who is trained to assess the impact of the stroke on the stroke survivor and their job; coordinate appropriate support from the National Health Service (NHS), employers and other stakeholders; negotiate workplace adjustments, monitor return-to-work, and explore alternatives where current work is not feasible or cannot be maintained [50]. ESSVR is delivered to community-dwelling stroke survivors in four stages (early recovery, graded return-to-work, job retention, and discharge), each comprising several core and desirable components. More information regarding the RETAKE trial and ESSVR can be found in the trial protocol [49].

### 2.3. Training for Occupational Therapists

OTs were invited to attend a two-day, manualised, face-to-face training session facilitated by expert trainers and mentors in VR. The aim of the session was to acquaint the OTs with the components of the intervention and the research process. Following their initial training, the OTs were also encouraged to attend monthly, hour-long mentoring sessions to receive ongoing mentoring support from an OT with expertise in VR. The aim of mentoring was to support the OTs to deliver the intervention with fidelity and to foster peer support through discussion of the OTs’ active ESSVR cases and sharing best practice. OTs were encouraged to contact their mentor outside of group mentoring if further support was needed. OTs attended a one-day, in-person, refresher training session six months after their initial training session.

### 2.4. Measures of OT Attributes

#### 2.4.1. Previous Experience and Knowledge

We designed a form to capture information about the OTs’ education levels, and years of experience in OT, stroke rehabilitation, and VR. The form also asked whether OTs had recent experience of health research (yes or no) and theoretical knowledge of VR (yes or no).

#### 2.4.2. Competency

OTs’ competency to deliver the intervention was assessed at three different timepoints: immediately following the two-day initial ESSVR training session, following the refresher training session held six months after training, and twelve months after the initial training session. At the initial and six-month competency assessments, the OTs were provided a vignette (that illustrated a case study) and the OT was asked to create a treatment plan. To assess competency at the third timepoint, OTs were asked to send completed intervention records from the first stroke survivor they treated nine months after their initial training session. This was performed to ensure that the therapists would theoretically be evaluated based on their treatment of a participant at their most experienced point of intervention delivery. In cases where OTs did not have nine months experience of delivering the intervention, the intervention records for their last treated stroke survivor were requested.

Competency was assessed by the central training team comprising two OTs with expertise in the ESSVR intervention and OT research. OT responses were marked against a rubric assessing their knowledge, clinical reasoning, and written communication. These scores were used to categorise competency as ‘needs support’, ‘competent’, and ‘highly competent’. Assessments were independently double-marked and any discrepancies that affected classification were discussed and agreed between the two raters.

#### 2.4.3. Engagement with Mentoring

The amount of contact each OT had with their mentor, inside and outside of their mentoring group, was recorded in minutes. Mentoring was summarised over two time periods, the amount of mentoring each OT received before their selected fidelity case began, and the amount of mentoring received during their selected fidelity case.

### 2.5. Fidelity Assessment

The fidelity assessment used a retrospective review of stroke survivor intervention records that included session content case report forms (CRFs), OT clinical notes, and correspondence between the OT, stroke survivors and other key stakeholders to assess intervention fidelity (see Table 1). Stroke survivor intervention records that were collected for competency assessment were also used for the fidelity assessment (see above). Once the stroke survivor intervention records were obtained, researchers (KP, JP) used an ESSVR-specific fidelity checklist and its associated guidance notes to assess component delivery. The total fidelity assessment score was calculated based on the number of components delivered divided by the total number of components that were deliverable and multiplied by 100 to provide a percentage of fidelity (0–100%). More information regarding the development and psychometric properties of the ESSVR fidelity checklist can be found elsewhere [51].

### 2.6. Return-to-Work Outcome

Stroke survivor RTW outcome was assessed 12 months post-randomisation. Stroke survivors were asked whether they had returned to work (yes or no). The definition of RTW in this study was “return to paid or unpaid work, for at least two hours per week”. This could include returning to the previous role or working in a new role.

### 2.7. Statistical Methods

A series of univariate linear regression analyses were performed to identify any potential predictors. Statistically significant results were adjusted for potential confounding factors. In cases where more than one variable was found to predict the fidelity score, a multivariate linear regression analysis was conducted.

## 3. Results

### 3.1. Description of Occupational Therapists

Data on therapist-level attributes were collected between February 2018 and November 2020. A total of 46 OTs across 16 sites were recruited and trained to deliver ESSVR. Of these, 39 OTs sent the stroke survivor intervention records as requested (one stroke survivor per OT, *n* = 39). Non-response was due to illness (*n* = 2) and no recruitment of ESSVR participants (*n* = 3), For demographic characteristics of the OTs see Table 2.

### 3.2. Fidelity Scores

Fidelity assessment score ranged from 30.8% to 100%, with an average score of 78.8% (*SD*: 19.2%).

### 3.3. OT Attributes

#### 3.3.1. Relationship between OT Attributes and Fidelity Assessment Score

Data regarding experience, post-training competence, engagement with mentoring, and fidelity assessment were collected and analysed for all 39 OTs. Of the nine OT attributes analysed through a series of simple linear regression calculations, only an average amount of mentoring received per month was a significant predictor of fidelity assessment score (*F*(1, 37) = 6.21, *p* < 0.05, with an *R*^2^ of 0.12). OTs’ predicted fidelity assessment score was equal to 67.86 + 0.29% (minutes of mentoring). Fidelity assessment score increased 0.29% for each minute of mentoring received per month. This effect remained significant when adjusted for potential confounding variables (experience, knowledge, and total previous amount of mentoring received).

See Table 3 for the individual relationships between OT attributes and fidelity assessment score.

#### 3.3.2. Relationships between OT Attributes and Stroke Survivor RTW Outcomes

Univariate logistic regression was conducted to explore and identify attributes that might be associated with the likelihood that stroke survivors would RTW following ESSVR delivery. Increase in years of stroke rehabilitation experience (OR = 1.16, 95% CI [1.02, 1.32]), increase in average minutes of mentoring received monthly (OR = 1.03, 95% CI [1.0, 1.07]), and fidelity assessment score (OR = 1.06, 95% CI [1.01, 1.1]) were found to be independently associated with increased likelihood of a stroke survivor’s RTW (See Table 4). The attributes shown to be independently associated with increased likelihood of RTW were included in a multivariate logistic analysis.

A multivariate logistic regression analysis was performed to further explore the relationship between OT stroke rehabilitation experience and fidelity of ESSVR delivery on the likelihood that stroke survivors would RTW following ESSVR. The logistic regression model was statistically significant [*X*^2^ (2, *N* = 39) = 14.07, *p* = 0.001]. The model explained 30.3% (Cox and Snell *R*^2^) of the of the variance in RTW outcome. Increasing years of stroke rehabilitation experience (OR = 1.17, 95% CI [1.02, 1.35]) and increasing fidelity assessment score (OR = 1.06, 95% CI [1.01, 1.1]) was associated with an increase in the likelihood of returning to work (See Table 5).

## 4. Discussion

Little evidence exists regarding what therapist attributes might impact fidelity and patient outcomes in complex rehabilitation interventions outside of the psychotherapy literature. This study found that only greater amounts of mentoring received per month during the stroke survivor case selected for fidelity assessment was associated with higher rates of fidelity to ESSVR delivery, and that more experience of stroke rehabilitation and higher fidelity rates were associated a greater likelihood of stroke survivor return-to-work at 12 months post-randomisation.

Despite the importance of assessing fidelity in clinical trials, it is a construct often overlooked in occupational therapy interventions. The OTs in the present study were able to deliver ESSVR with 78.8% fidelity on average, which is relatively high compared with other studies with similar evaluations [52,53]. These findings suggest that ESSVR was delivered with acceptable (70% or higher) rates of fidelity and that the wider RETAKE trial might not be impacted by issues of poor fidelity.

Engagement with mentoring is predictive of fidelity in interventions delivered by OTs in other studies [54]. The complexity and individualisation required by ESSVR in combination with the observation that the average amount of engagement per month, not the total amount of mentoring previously received was predictive of fidelity assessment score, might suggest that while mentoring does not necessarily develop OTs into experts in ESSVR, ongoing engagement with mentoring might support the OTs to deliver ESSVR with fidelity. This might be due to the structure of the mentoring in RETAKE, which created opportunity for further knowledge acquisition and peer support as well as supervision from an expert mentor [55,56]. These results, taken with the results of Döpp and colleagues’ [54] study, suggest that mentoring might be an effective implementation strategy for OTs delivering complex interventions. The finding that the total amount of mentoring received was not associated with fidelity is surprising, and future research should consider what further experience or training would be required to facilitate OTs reaching an ‘expert’ level of delivery in ESSVR. Regardless, future studies of complex interventions should consider the inclusion of a mentoring programme and encourage the therapists to engage with the programme to support fidelity of intervention delivery.

Higher rates of fidelity have long been associated with more positive patient treatment outcomes [25,57]. The results of the present study reinforce such findings. What is surprising is that the OTs’ previous experience of stroke rehabilitation was related to the return-to-work outcome, but not their previous experience of vocational rehabilitation. OTs with more years of experience in stroke rehabilitation maybe able to better understand the contextual factors that would prevent someone from returning to work after their stroke which in turn might lead to the OT individualising ESSVR in a way that more efficiently supports the stroke survivor to return-to-work. Further research is required to understand this relationship. Mentoring may have helped OTs with a wide range of VR experience to deliver ESSVR with fidelity. This suggests that consistent, timely mentoring support may be more important in the implementation of VR than previous experience of delivering it.

In our study, initial competence to deliver ESSVR was hypothesized to be a factor with the potential to influence fidelity outcomes because it has previously been demonstrated as a predictive factor of implementation fidelity in other studies [28,58]. Contrary to the previous evidence, therapist competence was not indicative of fidelity in this study. However, this discrepancy may be because the amount of time between the OTs’ initial competency assessment and the case sampled for fidelity assessment varied greatly, with most OTs having at least six months of experience before starting the intervention with the stroke survivor selected for fidelity assessment. Most OTs (*n* = 34; 87%) also attended their refresher training session. In the time elapsed between their initial training and their selected fidelity assessment case, the OTs built on their initial competence and understanding of the intervention. Future studies might investigate this further by measuring attributes at additional timepoints and exploring the changes in attributes over time that might occur due to involvement in clinical research and engagement with intervention training.

This study’s limitations should be seen in light of the exploratory nature of the study, coupled with the scarcity of other such occupational therapy and VR research attempting to explore the relationships between therapist attributes, fidelity, and treatment outcomes outside of psychotherapy. This is an important consideration because whilst psychotherapy is also a complex intervention, it is different from VR and is typically delivered by a non-occupational therapy workforce. Therefore, we cannot assume that the research findings from the field of psychotherapy necessarily translate/transfer to VR- and OT-delivered interventions.

This study included large numbers of potential predictor variables and limited numbers of OTs and stroke survivors. Because of the small sample size, we were unable to explore in depth the interactions between the predictor variables themselves. The small sample of OTs in this study mostly included women (which is representative of the national picture of the profession, with 91% of OTs being women) [59] with relatively little self-reported recent research experience. However, it is however difficult to generalise the results of the study to a larger population of OTs who might have more extensive research experience.

To assess fidelity in this study, OTs were asked for a specified stroke survivor’s intervention records to which a fidelity checklist [51] was applied. Fidelity checklist completion was dependent on the record keeping of the OTs and completeness of the intervention records, which limits fidelity conclusions to ‘evidence of’ the delivery of components. OTs were trained to maintain their intervention records in a way that was easily accessible for the research team; however, it is possible that OTs might not have recorded evidence of component delivery. Additionally, this study sampled one stroke survivor’s intervention records per OT, which begs the consideration that the cases sampled might not be a true reflection of the OTs’ actual overall fidelity. Future studies might look to examine several stroke survivors per OT and explore the changes in fidelity assessment score over time and the factors associated with those changes.

This study did not explore the impact of stroke survivor attributes on return-to-work outcome. There is a plethora of systematic reviews examining predictors of return-to-work after stroke which have identified attributes related to high likelihood of returning to work after a stroke, such as milder stroke severity [60], being male [61], and having independence in activities of daily living [61]. What is lacking from these systematic reviews, and from studies of return-to-work after stroke more generally, is greater consideration of the impact of work-related attributes (such as the adaptability of the stroke survivor’s role or the relationships with employers), which should be considered in future studies. The present study also did not consider the impact of organisational factors, which would have provided further context for the environment in which the OTs were delivering the intervention. For example, pressure for service development and organisational motivation to address the needs of a changing healthcare climate are factors that are associated with greater therapist fidelity rates [62]. Additionally, resource availability (i.e., adequate staffing, capacity, and service financial resources) might negatively impact fidelity and should be considered in future studies.

We suggest that providing support from expert mentors to OTs is a key implementation strategy for ensuring the faithful delivery of ESSVR and similar interventions. Future research should seek to further explore the mechanisms of action within mentoring to understand what underlying mechanisms of the mentoring might be facilitating delivery with fidelity (e.g., peer support, discussion of cases, etc.). Future studies should seek to include higher numbers of therapists and stroke survivors to achieve the statistical power needed to explore the relationships between attributes, fidelity, and patient outcomes more effectively [12,63,64].

## 5. Conclusions

Taken together, the findings of this study suggest that upskilling OTs to deliver complex interventions, such as ESSVR, in clinical trials may require mentoring support in addition to training to ensure fidelity. Furthermore, providing mentoring to ensure intervention fidelity may positively influence individual participant outcomes in return-to-work after stroke.

## Figures and Tables

**Table 1 ijerph-20-04694-t001:** Brief descriptions of the components of the stroke survivor intervention records.

Intervention Record Component	Description
Content CRFs	OTs indicate which components of the intervention and other common OT practices were delivered in a session.
Therapist clinical notes	OT notes from each instance of contact with the stroke survivor or other key stakeholders.
Supplementary material	Extra materials provided in the case file. Includes: Evidence of correspondence (e.g., copies of emails and written communication to key stakeholders); Educational information provided to key stakeholders.

**Table 2 ijerph-20-04694-t002:** Attributes of the 39 OTs delivering ESSVR.

Attribute	*n* (% of Sample)
Gender	
Female	35 (90%)
Male	4 (10%)
**Job Factors**	
Clinical Role ^a^	
OT	31 (79%)
OT Team Leader	4 (10%)
Therapy Manager	2 (5%)
Independent OT	1 (3%)
Senior Research Assistant	1 (3%)
NHS Band ^b^	
Band 6	24 (62%)
Band 7	15 (38%)
**Experience**	**Mean (Standard Deviation)**
Years qualified as OT	17.3 (7.95)
Years of experience:	
Stroke rehabilitation	9.34 (7.17)
VR	3.55 (4)
Recent research experience	
Yes	7 (18%)
No	32 (82%)
**Knowledge**	
Theoretical knowledge of VR	
Yes	22 (56%)
No	17 (44%)
Initial Competency Assessment	
Needs support	9 (23%)
Competent	28 (72%)
Highly competent	2 (5%)
**Engagement with Mentoring**	**Mean (Standard Deviation)**
Total minutes of mentoring received	378.74 (286.38)
Average minutes of mentoring per month	37.77 (25.02)

Abbreviations: NHS: National Health Service; OT: Occupational Therapist; VR: Vocational Rehabilitation. ^a^ Job titles of “OT”, “OT Team Leader”, and “Therapy Manager” represent increasing responsibility within the NHS. The one “Independent OT” was working in private practice and was linked to an NHS site for study purposes. The “Senior Research Assistant” was a qualified, practicing OT, but had a clinical research role within their institution. ^b^ The NHS band system allocates a point score to each role in the NHS, which determines the basic rate of salary for the role. The higher the band, the more pay and experience associated with the role. The typical entry-level band for OTs is Band 5.

**Table 3 ijerph-20-04694-t003:** Relationship between OT attributes and fidelity assessment score for 39 OTs.

Attributes	*β*	95% Confidence Interval		*p*
		Lower	Upper	
**Experience**				
Years qualified as OT	0.28	−0.52	1.07	0.49
Years of stroke rehabilitation experience	0.41	−0.48	1.28	0.36
Years of VR experience	−0.29	−1.89	1.31	0.72
**Knowledge**				
Level of education	1.61	−9.34	12.55	0.77
Theoretical knowledge of VR	6.95	−5.55	19.45	0.27
Recent research experience	11.14	−4.86	27.14	0.17
Initial Competence	6.44	−7.00	19.89	0.34
**Engagement**				
Amount of mentoring received pre-fidelity case (minutes)	0.01	−0.01	0.03	0.43
Average monthly amount of mentoring received (minutes)	0.29	0.05	0.53	0.02 *

Abbreviations: OT: Occupational Therapist; VR: Vocational Rehabilitation. * *p* < 0.05.

**Table 4 ijerph-20-04694-t004:** Relationship between OT attributes and stroke survivor RTW outcomes at 12 months post-randomisation explored by univariate logistic regression.

Attributes	Odds Ratio	95% Confidence Interval		*p*
		Lower	Upper	
**Experience**				
Years qualified	1.1	1.0	1.2	0.05
Years of stroke rehabilitation experience	1.16	1.02	1.32	0.02 *
Years of VR experience	1.19	0.98	1.45	0.08
**Knowledge**				
Level of education	1.58	0.51	4.92	0.43
Theoretical knowledge of VR	0.82	0.22	3.0	0.76
Recent research experience	0.59	0.1	3.49	0.56
Initial Competence	1.71	0.91	33.35	0.06
**Engagement**				
Amount of mentoring received pre-fidelity case (minutes)	1.0	1.0	1.0	0.16
Average monthly amount of mentoring received (minutes)	1.03	1.0	1.07	0.04 *
**Fidelity**				
Fidelity assessment score (%)	1.06	1.01	1.1	0.01 *

Abbreviations: OT: Occupational Therapist; VR: Vocational Rehabilitation. * *p* < 0.05.

**Table 5 ijerph-20-04694-t005:** Relationship between OT attributes and stroke survivor RTW outcomes at 12 months post-randomisation through univariate logistic regression.

Attributes	Odds Ratio	95% Confidence Interval		*p*
		Lower	Upper	
**Experience**				
Years of stroke rehabilitation experience	1.17	1.02	1.35	0.03 *
**Fidelity**				
Fidelity assessment score (%)	1.06	1.01	1.1	0.02 *

* *p* < 0.05.

## Data Availability

On study completion, the final trial dataset will be archived at the University of Nottingham. Following completion of the RETAKE trial and publication of its effectiveness outcomes, any party may apply to the corresponding author for access to the dataset. Access will be governed by an information governance committee formed between The University of Nottingham and the University of Leeds.

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
