# Peer review of "Exploring the Association between Individual-Level Attributes and Fidelity to a Vocational Rehabilitation Intervention within a Randomised Controlled Trial"

_ijerph, 2023, doi:10.3390/ijerph20064694_

Round 1

Reviewer 1 Report

Thank you for allowing me to review this manuscript. It has been of great interest to me. Please take my comments and questions as a way to improve the submitted manuscript.

Wouldn't it be advisable to include activities of daily living and work in keywords? Stroke?

Introduction

I would add the value of the occupational therapist in this field as a holistic professional dedicated to activities of daily living. I think this needs to be included in the discussion and conclusions as well.

The inclusion criteria should also specify how the ability to “provide informed consent” was measured.

I would talk about sex not gender

Reviewer 2 Report

Dear authors,

Congratulations on the study. I don't see any problems with the manuscript.

As a suggestion, I would remove the sentence on page 2, lines 52-53 (In this study, we only focus on the therapists’ adherence to the intervention part of fidelity, because therapist competence is addressed in a separate related study). I believe this information is unnecessary in the introduction and can be placed in the discussion when it is commented on regarding the limitations and future directions.

Reviewer 3 Report

The manuscript entitled “Exploring the association between individual-level attributes 2 and fidelity to a vocational rehabilitation intervention in a randomised controlled trial” explore the relationships between attributes of OTs, fidelity of ESSVR and stroke survivor RTW outcomes.

The topic is interesting and help clinicians to clarify possible factors that affecting stroke survivor’s return to work. Overall, the description of the study is not comprehensive. And I suggested the manuscript can be shortened in Introduction and discussion, to make it more condensed.

The following are suggestions:

Line 42-48

The paragraph aims to explain “complex” interventions, please shortened the sentences to be structured.

Line 52-54

The introduction is to give background information regarding the study purpose, note to explain what you are focused on. Please remove the description or move it to the Method

Line 49-68

These paragraphs are explaining “fidelity”, including definition, assessment and the importance of fidelity. I suggest rearrangement the paragraphs into one and being more concise.

Line 72

Evidence-base practice (EBP), the abbreviation shall be placed here not in the Line 79.

Line 103

Here the author mentioned this was a large multi-centre randomized controlled trial, but in the Material and Methods, I did not find the description of RCT, including randomization, allocation, blinding, etc.  Was this only a small part of analysis of a trial?  Please explain the details.

Also, the author used the title “…  in a randomized controlled trial”, seems to explain this is a part of a RCT. I would suggest revise the title, in order not to mislead the readers.  

Line 112 participants

Did the stroke survivors being allocated into different groups in the original design of the RCTs? Or were they all in the same group.

The inclusion criteria of stroke survivors was not clear

What were their consciousness level, motor, speech, sensory condition.

If the recurrent stroke patient being enrolled.

Line 127

Remove the comma between Early and stroke.

Line 132

“A more detailed description of the intervention can be found elsewhere [50]” This sentence is redundant, please remove it, and move the reference to the previous sentence.

Line 173

The paragraph was similar to those in Line 145-147

Line 194-196

How did the study define Return to Work? Were the stroke survivors necessary to back the their previous work or any kind?  What was the duration of working will be considered work? Part time job?

Results

Though the study focused on OTs, however, the information of how many stroke survivors received the vocational therapy is lacking. This is important.

Table 2

What is difference between Staff OT, Independent OT, Therapy Manager?

M (SD), please use Mean (Standard deviation), no need of abbreviation

Table 4

The data of  “Initial Competence” is lacking, please make sure the data were collect during the typesetting of table.

Line 267-269

What percentage of fidelity did the author consider acceptable?

Line 299-300

“In our study, competence to deliver ESSVR was considered a factor with the potential to influence fidelity outcomes” However, the results did not suggest the competence related to the fidelity. Please explain your meaning.

Line 312-319 and Line 339-353

These two paragraphs describe the limitation of the present study. However, the paragraph (Line 328-338) was not a part of limitation.

And the last paragraph Line 354-363 seems too redundant, please make it concise.

Round 2

Reviewer 3 Report

Thanks for the revision. I have no more comment.